# Lower Complexity Bounds for Nonconvex-Strongly-Convex Bilevel Optimization with First-Order Oracles

Kaiyi Ji [1]

## Abstract

Although upper bound guarantees for bilevel optimization have been widely studied, progress on lower bounds has been limited due to the complexity of the bilevel structure. In this work, we focus on the smooth nonconvex-strongly-convex setting and develop new hard instances that yield nontrivial lower bounds under deterministic and stochastic first-order oracle models. In the deterministic case, we prove that any first-order zero-respecting algorithm requires at least $\Omega(\kappa^{3/2}\epsilon^{-2})$ oracle calls to find an $\epsilon$-accurate stationary point, improving the optimal lower bounds known for single-level nonconvex optimization and for nonconvex-strongly-convex min-max problems. In the stochastic case, we show that at least $\Omega(\kappa^{5/2}\epsilon^{-4})$ stochastic oracle calls are necessary, again strengthening the best known bounds in related settings. Our results expose substantial gaps between current upper and lower bounds for bilevel optimization and suggest that even simplified regimes, such as those with quadratic lower-level objectives, warrant further investigation toward understanding the optimal complexity of bilevel optimization under standard first-order oracles.

## 1. Introduction

In this paper, we are interested in solving the following bilevel optimization problem:

$$\min_{x \in \mathcal{X}} H(\mathbf{x}) := f(\mathbf{x}; \mathbf{y}^*(\mathbf{x}))$$
$$\text{s.t. } \mathbf{y}^*(\mathbf{x}) = \arg\min_{\mathbf{y} \in \mathcal{Y}} g(\mathbf{x}; \mathbf{y}), \qquad (1)$$

[1]Department of Computer Science and Engineering, University at Buffalo, New York, United States. Correspondence to: Kaiyi Ji <kaiyiji@buffalo.edu>.

*Proceedings of the 43$^{rd}$ International Conference on Machine Learning*, Seoul, South Korea. PMLR 306, 2026. Copyright 2026 by the author(s).

where $\mathcal{X} \subset \mathbb{R}^m$ and $\mathcal{Y} \subset \mathbb{R}^n$ are nonempty closed convex sets. In this paper, we study the smooth nonconvex-strongly-convex bilevel optimization setting, where the lower-level function $g$ is smooth and strongly convex in $\mathbf{y}$, while the upper-level function $f$ is smooth and potentially nonconvex. This formulation captures a variety of modern applications, including meta-learning (Rajeswaran et al., 2019), reinforcement learning (Konda & Tsitsiklis, 2000; Hong et al., 2023), robotics (Wang et al., 2024), as well as communication networks and federated learning (Ji & Ying, 2023; Tarzanagh et al., 2022; Huang et al., 2023).

Recent years have witnessed substantial progress in understanding the convergence and complexity of bilevel optimization. A broad class of works (Ji et al., 2021; Hong et al., 2023; Chen et al., 2022; Dagréou et al., 2022) analyzes nonconvex-strongly-convex bilevel problems under access to second-order information such as Hessian- and Jacobian-vector products. More recently, there has been growing interest in developing and analyzing *fully first-order* bilevel algorithms that avoid any second-order computations (Shen & Chen, 2023; Chen et al., 2025; Lu & Mei, 2024; Kwon et al., 2023; Liu et al., 2020).

Although upper-bound complexity analyses for bilevel optimization have been extensively studied, progress on establishing tight *lower bounds* has been much slower. This difficulty largely stems from the intrinsic complexity of the general bilevel formulation. In particular, deriving meaningful lower bounds that capture the dependence on condition numbers and the target accuracy $\epsilon$ requires carefully constructed hard instances; otherwise, one risks obtaining vacuous bounds that are no stronger than classical single-level lower bounds. Ji & Liang (2023) establish lower bounds for strongly-convex–strongly-convex and convex–strongly-convex bilevel problems under second-order oracle access, assuming that the hyper-objective $H(\mathbf{x})$ is convex or strongly convex. Their results reveal a gap of a factor $\sqrt{\kappa}$ compared to corresponding lower bounds for min–max optimization under analogous assumptions, where $\kappa$ denotes the condition number of the lower-level function. However, their analysis is limited to the deterministic setting, and the convexity assumptions on the hyper-objective may be restrictive for general bilevel problems. Dagréou et al. 2024 derive a lower bound of $\Omega(n + \sqrt{n}\,\epsilon^{-2})$ for finite-

sum nonconvex–strongly-convex bilevel problems. This bound, however, does not reflect the dependence on condition numbers and is weaker than the known lower bounds for min-max problems of the same type (Zhang et al., 2021). More recently, Kwon et al. (2024) establish lower bounds for nonconvex–strongly-convex bilevel optimization under a so-called $\mathbf{y}^*$-aware stochastic first-order oracle, which returns an estimate $\hat{\mathbf{y}}$ that is $\epsilon$-close to the exact lower-level solution $\mathbf{y}^*$. This oracle effectively reduces the problem to one resembling single-level optimization. Nevertheless, lower bounds for standard (stochastic) first-order oracles that directly access the upper- and lower-level functions $f$ and $g$ remain an open problem.

In this paper, we take a further step toward reducing this gap by developing nontrivial lower bounds for smooth nonconvex–strongly-convex bilevel optimization under standard first-order oracle models. Our main contributions are summarized below.

- **Deterministic setting.** We construct a hard instance on which no first-order zero-respecting algorithm can find an $\epsilon$-stationary solution using fewer than $\Omega(\kappa^{3/2}\epsilon^{-2})$ first-order oracle calls for smooth nonconvex-strongly-convex bilevel problems. In comparison, the optimal lower bounds for related settings are $\Omega(\epsilon^{-2})$ for general smooth nonconvex single-level optimization (Carmon et al., 2020) and $\Omega(\sqrt{\kappa}\epsilon^{-2})$ for smooth nonconvex-strongly-convex min–max optimization (Li et al., 2021). Our result improves these bounds by factors of $\kappa^{3/2}$ and $\kappa$, respectively.

  On the upper-bound side, Chen et al. (2025) propose a first-order penalty method achieving a convergence rate of order $\kappa^4\epsilon^{-2}$, which can be reduced to $\kappa^{3.5}\epsilon^{-2}$ through a naive application of Nesterov acceleration. However, even when compared with our lower bound, there remains a gap of order $\kappa^2$, indicating substantial room for future improvements.

- **Stochastic setting.** We further construct an instance showing that no first-order zero-respecting algorithm can achieve an $\epsilon$-stationary solution with fewer than $\Omega(\kappa^{5/2}\epsilon^{-4})$ stochastic oracle calls under bounded variance assumptions. For comparison, the lower bound for standard smooth nonconvex single-level stochastic optimization is $\Omega(\epsilon^{-4})$ (Arjevani et al., 2023), and for smooth nonconvex–strongly-convex min–max optimization it is $\Omega(\kappa^{1/3}\epsilon^{-2})$ (Li et al., 2021). Our result improves upon these by factors of $\kappa^{5/2}$ and $\kappa^{13/6}$, respectively. Compared with the $\Omega(\epsilon^{-6})$ upper bound established by Kwon et al. (2024), a gap still remains.

- **Implications.** Our constructions demonstrate that nontrivial lower bounds for nonconvex-strongly-convex bilevel optimization are indeed possible and are significantly stronger than the known results for single-level

and min–max problems. Nevertheless, substantial gaps persist between current upper and lower bounds, even in this restricted setting. Motivated by our findings, we suggest that closing these gaps may require first studying the simpler yet meaningful case in which the lower-level function is **quadratic**. Our lower bounds continue to apply in that regime, but obtaining tighter upper bounds in this setting remains largely unexplored and not yet well understood. We hope that the results presented in this paper offer valuable insights for future progress in this direction.

## 2. Related Works

**Bilevel optimization algorithms.** Bilevel optimization has a long history dating back to the seminal work of Bracken & McGill (1973). Early studies (Hansen et al., 1992; Shi et al., 2005) approached bilevel programs from a constrained optimization perspective, motivating the development of KKT-based reformulations and related techniques. More recently, gradient-based bilevel optimization has attracted significant attention due to its efficiency and scalability in modern machine learning applications. A major class of gradient-based approaches is the family of Approximate Implicit Differentiation (AID) methods (Domke, 2012; Liao et al., 2018; Ji et al., 2021; Dagréou et al., 2022; Yang et al., 2024), which compute the hypergradient via implicit differentiation and approximate the resulting linear system using iterative solvers. In contrast, Iterative Differentiation (ITD) methods (Maclaurin et al., 2015; Franceschi et al., 2017) estimate hypergradients by unrolling the lower-level optimization and applying automatic differentiation in either forward or reverse mode. Building upon these ideas, a number of stochastic bilevel algorithms have been developed using Neumann-series approximation (Chen et al., 2022; Ji et al., 2021), recursive momentum techniques (Yang et al., 2021; Guo & Yang, 2021), and variance-reduction mechanisms (Yang et al., 2021). All such methods rely on second-order information, commonly in the form of Hessian–vector or Jacobian–vector products. A comprehensive overview is provided in the survey (Liu et al., 2021a).

Recently, growing interest has shifted slightly toward designing *first-order* bilevel optimization methods that use only (stochastic) first-order oracles, thereby avoiding explicit second-order computations. Representative examples include penalty-based methods (Shen & Chen, 2023; Lu & Mei, 2024; Kwon et al., 2023; Jiang et al., 2025; Chen et al., 2025), primal–dual frameworks (Sow et al., 2022), finite-difference Hessian–vector approximation techniques (Yang et al., 2023), value-function-based approaches (Liu et al., 2020; 2021c;b), barrier-based formulations (Liu et al., 2022), and min–max optimization based methods (Lu & Mei, 2026; Wang et al., 2023). These works collectively highlight the potential of first-order bilevel algorithms to

achieve competitive performance while significantly reducing computational overhead.

**Upper bound analysis.** A large body of work, including Ji et al. 2021; Hong et al. 2023; Chen et al. 2022, studies AID- and ITD-type algorithms for nonconvex–strongly-convex bilevel optimization. Another line of research considers cases where the lower-level objective is not strongly convex; for example, Arbel & Mairal 2022; Liu et al. 2021c analyze settings in which the lower-level solution is characterized through a selection map (e.g., the output of a particular algorithm). For bilevel algorithms that rely solely on (stochastic) first-order oracles, Kwon et al. 2023; Chen et al. 2025 establish convergence guarantees for nonconvex–strongly-convex formulations. Shen & Chen 2023; Chen et al. 2024 study algorithms under weaker structural assumptions on the lower-level problem, extending beyond strong convexity.

**Lower bound analysis.** Foundational lower bounds for first-order optimization were established by Nemirovski and Nesterov and are presented in their textbooks (Nemirovsky, 1992; Nesterov et al., 2018). A central concept in this theory is the notion of *zero-chains*, which ensure that any zero-respecting first-order method can activate coordinates only sequentially. Recent works have significantly advanced these constructions in the context of smooth nonconvex optimization (Fang et al., 2018; Carmon et al., 2020; 2021; Arjevani et al., 2023). Building upon these developments, Li et al. (2021) establish lower bounds for nonconvex–strongly-convex min–max optimization. Our work is highly inspired by these results.

Lower bounds for bilevel optimization are relatively underexplored. Ji & Liang (2023) derive bounds for convex and strongly-convex bilevel problems using second-order oracles. More recently, Kwon et al. (2024) establish lower bounds for nonconvex-strongly-convex bilevel problems under a $\mathbf{y}^*$-aware stochastic oracle. Dagréou et al. 2024 derive a lower bound for finite-sum nonconvex-strongly-convex bilevel problems. In contrast, we provide lower bounds for nonconvex-strongly-convex bilevel optimization using standard (stochastic) first-order oracles.

**A concurrent work.** As we were preparing the final draft of this paper, we became aware of a concurrent nice work by Chen & Zhang (2025), which was posted on arXiv. This work also establishes lower bounds for nonconvex-strongly-convex bilevel optimization under (stochastic) first-order oracle access, showing a larger dependence on the condition number than those of min-max and single-level minimization problems of the same type. Despite addressing a similar question, the constructions in the two works differ substantially. For example, the construction introduces an additional auxiliary variable $z$, whereas our construction is simpler without such variable. In the stochastic set-

ting, Chen & Zhang (2025) eliminate the coupling variable $\mathbf{y}$ to reduce the dimensionality, while we instead use two bounded hypercubes to control the noise variances.

# 3. Preliminaries

**Notations.** We use bold lower-case letters to denote vectors and regular lower-case letters to denote scalars. For a vector $\mathbf{x} \in \mathbb{R}^d$, we use $\mathbf{x}^t$ to denote its value at the $t^{th}$ iteration, and $x_i$ to denote its $i$th coordinate and define its support as $\mathrm{supp}(\mathbf{x}) := \{ i \mid x_i \neq 0 \}$. We use $\|\mathbf{x}\|_2 = \sqrt{\sum_{i=1}^d x_i^2}$ and $\|\mathbf{x}\|_\infty = \max_{1 \leq i \leq d} |x_i|$ to denote the $\ell_2$ and $\ell_\infty$ norms, respectively. For a matrix $M \in \mathbb{R}^{m \times n}$, we use $M_{i,j}$ to denote its $(i,j)$th entry. We use $\|M\|_\infty = \max_{1 \leq i \leq m} \sum_{j=1}^n |M_{i,j}|$ for the matrix infinity norm and $\|M\|_2$ for its spectral norm. For a square matrix $M$, we let $\mathrm{diag}_m(M)$ denote the block diagonal matrix with $m$ identical copies of $M$ on the diagonal. We use standard asymptotic notation $\mathcal{O}(\cdot)$, $\Omega(\cdot)$, and $\Theta(\cdot)$.

## 3.1. Function Class

In this paper, we focus on the class of smooth nonconvex-strongly-convex bilevel problems that satisfy the standard assumptions used in first order bilevel optimization.

**Definition 1.** *Given $L_f, L_g \geq \mu > 0$, $C \geq 0$ and $\Delta > 0$, define $\mathcal{F}(L_f, L_g, \mu, \Delta)$ to be the set of function pairs $\{f, g\}$ such that $f : \mathcal{X} \times \mathcal{Y} \to \mathbb{R}$ and $g : \mathcal{X} \times \mathcal{Y} \to \mathbb{R}$ for some nonempty closed convex sets $\mathcal{X} \subset \mathbb{R}^m$ and $\mathcal{Y} \subset \mathbb{R}^n$ for all $m, n \in \mathbb{N}$, which satisfy the following assumptions:*

1. *Functions $f, g$ are continuously differentiable, $L_f$ and $L_g$-smooth respectively, jointly in $(\mathbf{x}, \mathbf{y})$ over $\mathcal{X} \times \mathcal{Y}$.*

2. *For every $(\mathbf{x}, \mathbf{y}) \in \mathcal{X} \times \mathcal{Y}$, there exists a numerical constant $C \geq 0$ such that $\|\nabla_y f(\mathbf{x}, \mathbf{y})\|_2 \leq C$.*

3. *For every $\mathbf{x} \in \mathcal{X}$, $g(\mathbf{x}, \cdot)$ is $\mu$-strongly-convex in $\mathbf{y}$, that is, for any $\mathbf{y}_1, \mathbf{y}_2 \in \mathcal{Y}$,*

$$g(\mathbf{x}; \mathbf{y}_1) \geq g(\mathbf{x}; \mathbf{y}_2) + \langle \nabla_{\mathbf{y}} g(\mathbf{x}; \mathbf{y}_2), \mathbf{y}_1 - \mathbf{y}_2 \rangle$$
$$+ \frac{\mu}{2} \|\mathbf{y}_1 - \mathbf{y}_2\|_2^2.$$

4. *There exists a numerical constant $\rho \geq 0$ such that the second-order derivatives $\nabla_{\mathbf{x}, \mathbf{y}}^2 g$ and $\nabla_{\mathbf{y}, \mathbf{y}}^2 g$ are well-defined and $\rho$-Lipschitz jointly in $(\mathbf{x}, \mathbf{y})$ for all $(\mathbf{x}, \mathbf{y}) \in \mathcal{X} \times \mathcal{Y}$.*

5. *$H(\mathbf{0}) - \min_{\mathbf{x} \in \mathcal{X}} H(\mathbf{x}) \leq \Delta$, where $H(\mathbf{x}) := f(\mathbf{x}; \mathbf{y}^*(x))$ is the hyper-objective function.*

*Note that for items 2 and 4, we only require the existence of numerical constants $C, \rho = \mathcal{O}(1)$.*

Although we do not explicitly specify $L_g$ and $L_f$ in this paper, we are primarily interested in the regime where these constants are independent of the strong convexity parameter $\mu$ and the target accuracy $\epsilon$.

### 3.2. Algorithm Class

We focus on algorithms that solve bilevel optimization problems using (stochastic) first order oracles. For clarity of presentation, we first define the (stochastic) first-order oracles considered in this work.

**Definition 2** (Deterministic first-order oracle). *The deterministic first-order oracle of a differentiable function $f : \mathcal{X} \to \mathbb{R}$ is a mapping $O : \mathbf{x} \mapsto (f(\mathbf{x}), \nabla f(\mathbf{x}))$ for $\mathbf{x} \in \mathcal{X}$.*

**Definition 3** (Stochastic first-order oracle). *The stochastic first-order oracle of a differentiable function $f : \mathcal{X} \to \mathbb{R}$ is a mapping $O : \mathbf{x} \mapsto (f(\mathbf{x}), G_f(\mathbf{x}; \xi))$ for $\mathbf{x} \in \mathcal{X}$, where $\xi$ is a random variable satisfying $\mathbb{E}_\xi [G_f(\mathbf{x}; \xi)] = \nabla f(\mathbf{x})$ and $\mathbb{E}_\xi \|G_f(\mathbf{x}; \xi) - \nabla f(\mathbf{x})\|_2^2 \leq \sigma_f^2$.*

Note that the algorithms rely on first-order oracles for both the upper- and lower-level objectives $f$ and $g$. In the stochastic setting, we assume for simplicity that the variances of the stochastic first-order oracles are identical, i.e., $\sigma_f = \sigma_g = \sigma$. We further focus on first-order bilevel algorithms that satisfy the following zero-respecting property:

**Definition 4** (Algorithm class). *For upper- and lower-level objective functions $f : \mathcal{X} \times \mathcal{Y} \to \mathbb{R}$ and $g : \mathcal{X} \times \mathcal{Y} \to \mathbb{R}$ and their first-order oracles $O_f : (\mathbf{x}, \mathbf{y}) \mapsto (f(\mathbf{x}; \mathbf{y}), \nabla f(\mathbf{x}; \mathbf{y}))$ and $O_g : (\mathbf{x}, \mathbf{y}) \mapsto (g(\mathbf{x}; \mathbf{y}), \nabla g(\mathbf{x}; \mathbf{y}))$, the $(t + 1)$-th iterate $(\mathbf{x}^{t+1}, \mathbf{y}^{t+1})$ satisfies:*

$$\mathbf{x}^{t+1} \in \Big\{ \mathcal{P}_\mathcal{X}(\mathbf{u}) : \mathrm{supp}(\mathbf{u}) \subset \bigcup_{0 \leq i \leq t} (\mathrm{supp}(\mathbf{x}^i) \cup$$
$$\mathrm{supp}(\nabla_\mathbf{x} f(\mathbf{x}^i; \mathbf{y}^i)) \cup \mathrm{supp}(\nabla_\mathbf{x} g(\mathbf{x}^i; \mathbf{y}^i)) \Big\};$$

$$\mathbf{y}^{t+1} \in \Big\{ \mathcal{P}_\mathcal{Y}(\mathbf{v}) : \mathrm{supp}(\mathbf{v}) \subset \bigcup_{0 \leq i \leq t} (\mathrm{supp}(\mathbf{y}^i) \cup$$
$$\mathrm{supp}(\nabla_\mathbf{y} f(\mathbf{x}^i; \mathbf{y}^i)) \cup \mathrm{supp}(\nabla_\mathbf{y} g(\mathbf{x}^i; \mathbf{y}^i)) \Big\}. \quad (2)$$

*A similar definition applies in the stochastic setting, where the gradients $\nabla f$ and $\nabla g$ are replaced by their corresponding stochastic first-order oracles.*

Note that the subspaces defined in Equation (2) permit both simultaneous and alternating updates of $\mathbf{x}$ and $\mathbf{y}$, thereby including single-loop and double-loop bilevel optimization algorithms. Consequently, the algorithm class introduced in Definition 4 covers all existing first-order bilevel optimization methods, including but not limited to penalty-based approaches (Shen & Chen, 2023; Lu & Mei, 2024),

primal–dual methods (Sow et al., 2022), finite-difference Hessian–vector–approximation methods (Yang et al., 2023), value-function-based approaches (Liu et al., 2020; 2021c;b), and barrier-based methods (Liu et al., 2022).

## 4. Lower Bounds in Deterministic Setting

### 4.1. Useful Techniques for Lower-Bound Construction

In this paper, we focus on the bilevel optimization setting where the lower-level function $g(\mathbf{x}; \mathbf{y})$ is strongly convex in $\mathbf{y}$, while the upper-level function $f(\mathbf{x}; \mathbf{y})$ is smooth and possibly nonconvex. For this reason, our constructions draw on key techniques and components from the worst-case instances of Nesterov et al. 2018 for smooth strongly convex functions and Carmon et al. 2020 for smooth nonconvex functions. Their core idea is to make sure their instances satisfy the following notion of zero-chain property:

**Definition 5** (Zero-chain). *A function $f : \mathcal{X} \subset \mathbb{R}^d \to \mathbb{R}$ is a (first-order) zero-chain if for every $1 \leq i \leq d$,*

$$\mathrm{supp}(\mathbf{x}) := \{i : x_i \neq 0\} \subset \{1, \ldots, i - 1\}$$
$$\implies \mathrm{supp}(\nabla f(\mathbf{x})) \subset \{1, \ldots, i\}.$$

Consider running a first-order algorithm on a zero-chain function, starting from the initialization $\mathbf{x} = 0$, and assume access to a deterministic first-order oracle. By the zero-chain property, each iteration can introduce at most one new nonzero coordinate of $\mathbf{x}$—that is, each iteration "activates" at most one additional coordinate. Consequently, after $t$ iterations we must have $\mathrm{supp}(\mathbf{x}^t) \subset \{1, \ldots, t\}$. Therefore, if a good solution requires that at least $T$ coordinates be discovered, then any deterministic first-order method must take at least $T$ iterations, which yields a lower bound of order $T$ on the algorithm's complexity.

Following this strategy, Nesterov et al. 2018 and Carmon et al. 2020 provide the following key components for their constructions in strongly-convex and nonconvex settings:

- **Tri-diagonal matrix** $A$. Following Nesterov et al. 2018; Li et al. 2021, we use the following tri-diagonal 1-D discrete Laplacian matrix $A \in \mathbb{R}^{n \times n}$ to construct the strongly-convex lower-level instance:

$$A := \begin{bmatrix} 1 & -1 & & & \\ -1 & 2 & -1 & & \\ & \ddots & \ddots & \ddots & \\ & & -1 & 2 & -1 \\ & & & -1 & 1 \end{bmatrix}, \quad (3)$$

where it is verified that $A$ is positive semidefinite and $\|A\|_2 \leq 4$. Due to its tri-diagonal nature, it is easily verified that if $\mathrm{supp}(x) \subset 1, ..., i - 1$, then $Ax \subset \{1, ..., i\}$. In other words, if a vector has nonzero entries only at its first $i - 1$ coordinates, then multiplying it by $A$ can activate at most one additional coordinate, namely the $i$-th one.

- $\Psi(\cdot)$ **and** $\Phi(\cdot)$ **hardness functions.** Following the construction in Carmon et al. 2020, we employ the component functions $\Psi(x) : \mathbb{R} \to \mathbb{R}$ and $\Phi(x) : \mathbb{R} \to \mathbb{R}$ defined below.

$$\Psi(x) := \begin{cases} 0, & x \le \frac{1}{2}, \\ \exp\left(1 - \frac{1}{(2x-1)^2}\right), & x > \frac{1}{2}, \end{cases}$$

$$\Phi(x) := \sqrt{e} \int_{-\infty}^{x} e^{-\frac{1}{2}t^2} \, dt, \tag{4}$$

which have the following key properties that will be used in our analysis.

**Lemma 1** (Carmon et al. 2020, Lemma 1). *The functions $\Phi$ and $\Psi$ satisfy*

1. *For all $x \le \frac{1}{2}$ and $k \in \mathbb{N}$, we have $\Psi^{(k)}(x) = 0$, where $\Psi^{(k)}$ denotes the $k^{th}$-order derivative.*
2. *For all $x \ge 1, |y| < 1$, we have $\Psi(x)\,\Phi'(y) > 1$.*
3. *Both $\Psi$ and $\Phi$ are infinitely differentiable. For all $k \in \mathbb{N}$, it holds that*

$$\sup_x \left|\Psi^{(k)}(x)\right| \le \exp\left(\frac{5k}{2}\log(4k)\right)$$

$$\sup_x \left|\Phi^{(k)}(x)\right| \le \exp\left(\frac{3k}{2}\log\frac{3k}{2}\right).$$

4. *The functions and derivatives $\Psi$, $\Psi'$, $\Phi$, $\Phi'$ are nonnegative and bounded, with $0 < \Psi < e, 0 < \Psi' < \sqrt{\frac{54}{e}}, 0 < \Phi < \sqrt{2\pi e}, 0 < \Phi' < \sqrt{e}$.*

Carmon et al. 2020 use a construction of $f(\mathbf{x}) = \sum_i \left[\Psi(-x_{i-1})\Phi(-x_i) - \Psi(x_{i-1})\Phi(x_i)\right]$, which together with $\Psi'(0) = \Psi(0) = 0$, ensures the zero-chain property that if $\mathbf{x} \subset \{1, ..., i-1\}$, then $\nabla f(\mathbf{x}) \subset \{1, ..., i\}$. Furthermore, as we will show later, the boundedness of $\Psi$, $\Psi'$, $\Phi$, and $\Phi'$ is crucial for constructing a valid worst-case instance within the bilevel class $\mathcal{F}(L_f, L_g, \mu, C, \Delta)$.

## 4.2. Main Result: A Lower Bound on First-Order Oracle Complexity

The following theorem establishes a complexity lower bound for deterministic first-order bilevel algorithms.

**Theorem 4.1.** *For any $L_f, L_g, \mu, \Delta, \epsilon > 0$ satisfying $\kappa = L_g/\mu \ge 1$ and $\frac{\Delta}{L_f} = \mathcal{O}(1)$, there exist functions $f : \mathbb{R}^m \times \mathbb{R}^n \to \mathbb{R}$ and $g : \mathbb{R}^m \times \mathbb{R}^n \to \mathbb{R}$ such that $\{f, g\} \in \mathcal{F}(L_f, L_g, \mu, \Delta)$ for some $m, n \in \mathbb{N}$ with their deterministic first-order oracles. For any first-order bilevel algorithm of the form in Definition 4, in order to find an $\epsilon$-accurate stationary point $\mathbf{x}$ such that $\|\nabla H(\mathbf{x})\|_2 < \epsilon$, the algorithm must use at least $\frac{C_0 \Delta L_f \kappa^{3/2}}{\epsilon^2}$ oracle calls, where $H(\mathbf{x}) = f(\mathbf{x}; y^*(\mathbf{x}))$ with $y^*(\mathbf{x}) = \arg\min_{\mathbf{y}} g(\mathbf{x}; \mathbf{y})$ is the hyper-objective, and $C_0$ is a numerical constant.*

Carmon et al. 2020 establish a lower bound of $\Omega(1/\epsilon^2)$ for smooth nonconvex optimization, and (Li et al., 2021) proves a lower bound of $\Omega(\sqrt{\kappa}/\epsilon^2)$ for smooth nonconvex-strongly-concave min-max optimization. Both results can be viewed as special cases of smooth nonconvex-strongly-convex bilevel optimization, for which we obtain in Theorem 4.1 a much larger lower bound of $\Omega(\kappa^{3/2}/\epsilon^2)$. This demonstrates that bilevel optimization is provably more challenging than min-max optimization. This observation is consistent with the fundamental hardness comparison for smooth strongly-convex–strongly-convex bilevel problems established in Ji & Liang 2023.

### 4.3. Proof Outline for Deterministic Lower Bound

We consider the following worst-case instance. For notational simplicity, define $x_0 \equiv \frac{\lambda}{C_l M_{n,n}}$.

$$f(\mathbf{x}; \widetilde{\mathbf{y}}) = \sum_{i=1}^{T} \frac{\lambda^2 L_f}{L} \left[ \Psi\left(-\frac{C_l}{\lambda} y_n^{(i-1)}\right) \Phi\left(-\frac{C_r}{\lambda} y_1^{(i)}\right) \right.$$
$$\left. - \Psi\left(\frac{C_l}{\lambda} y_n^{(i-1)}\right) \Phi\left(\frac{C_r}{\lambda} y_1^{(i)}\right) \right]$$

$$g(\mathbf{x}; \widetilde{\mathbf{y}}) = \sum_{i=0}^{T} \left[ \frac{L_g n^2}{2(4n^2+1)} (\mathbf{y}^{(i)})^\top \left(\frac{1}{n^2} I_n + A\right) \mathbf{y}^{(i)} \right.$$
$$\left. - L_g (\mathbf{b}_x^{(i)})^\top \mathbf{y}^{(i)} \right], \tag{5}$$

where $\mathbf{x} = [x_1, ..., x_T] \in \mathbb{R}^T$ is the upper-level variable, $\widetilde{\mathbf{y}} = [\mathbf{y}^{(0)}, \mathbf{y}^{(1)}, ...., \mathbf{y}^{(T)}]$ with each $\mathbf{y}^{(i)} \in \mathbb{R}^n$ is the lower-level variable, $y_j^{(i)}$ returns the $j^{th}$ coordinate of $\mathbf{y}^{(i)}$, and the dimension $n = \lfloor \sqrt{\frac{L_g - \mu}{4\mu}} \rfloor$, and the design of $\mathbf{b}_x^{(i)}$ is most critical, which is given by

$$\mathbf{b}_x^{(i)} = [0, 0, ...., x_i] = x_i \mathbf{e}_n,$$

where $\mathbf{e}_i$ denotes the $i^{\text{th}}$ standard basis vector, whose sole nonzero entry equals 1. For simple presentation, the numerical constants $C_l, C_r, L$ and the parameter $\lambda$ will be specified at a later stage.

**Validation of our constructed instance.** We first verify that our constructed instance belongs to the function class $\mathcal{F}(L_f, L_g, \mu, \Delta)$.

1. First, we need to verify $g(\mathbf{x}; \cdot)$ is $\mu$-strongly convex. Since the matrix $A$ is positive semidefinite, it can be verified that $\nabla^2 g(\mathbf{x}; \cdot) = \frac{L_g n^2}{4n^2+1} \text{diag}_{T+1}(A + \frac{1}{n^2})$. Let $M := \text{diag}_{T+1}\{A\}$. For any vector $z \in \mathbb{R}^{n(T+1)}$, write it as a block vector $\mathbf{z} = [\mathbf{z}_1, \mathbf{z}_2, ..., \mathbf{z}_m], \mathbf{z}_i \in \mathbb{R}^n$, we have $\mathbf{z}^\top M \mathbf{z} = \sum_{i=1}^{m} \mathbf{z}_i^\top A \mathbf{z}_i$. Since $A$ is positive semidefinite, each term $\mathbf{z}_i^\top A \mathbf{z}_i \ge 0$, so the sum is nonnegative. Hence $\mathbf{z}^\top M \mathbf{z} \ge 0$ for all $\mathbf{z}$, and therefore $M$ is positive semidefinite. This further implies that

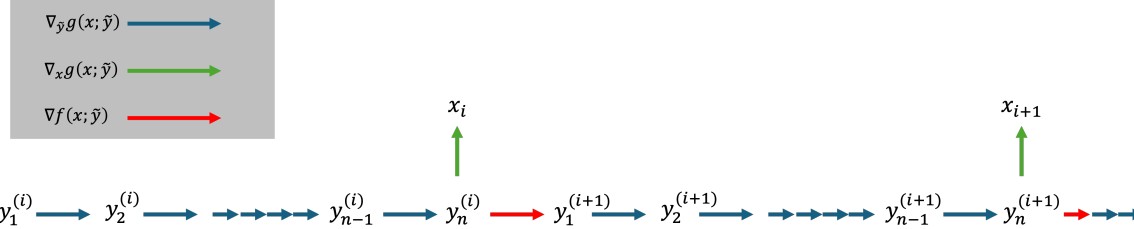

*Figure 1.* An illustration of the zero-chain for our constructed instance in Equation (5) for nonconvex-strongly-convex bilevel optimization.

$\|\nabla^2 g(\mathbf{x}; \cdot)\|_2 \geq \frac{L_g}{4n^2+1}$. Given that $n = \lfloor \sqrt{\frac{L_g - \mu}{4\mu}} \rfloor \leq \sqrt{\frac{L_g - \mu}{4\mu}}$, we have $\frac{L_g}{4n^2+1} \geq \mu$. This validates that $g(\mathbf{x}; \widetilde{\mathbf{y}})$ is $\mu$-strongly convex in $\widetilde{\mathbf{y}}$.

2. Next, we validate the smoothness of $f$ and $g$ functions:

   - For the lower-level function $g(\mathbf{x}; \widetilde{\mathbf{y}})$, it follows from Equation (5) that $\|\nabla^2_{\widetilde{\mathbf{y}}} g(\mathbf{x}; \widetilde{\mathbf{y}})\|_2 \leq \frac{L_g n^2}{4n^2+1}(\frac{1}{n^2} + 4) = L_g$, $\|\nabla^2_{\mathbf{x}, \widetilde{\mathbf{y}}} g(\mathbf{x}; \widetilde{\mathbf{y}})\|_2 = L_g, \nabla^2_{\mathbf{x}} g(\mathbf{x}; \widetilde{\mathbf{y}}) = 0$ for any $\mathbf{x}, \widetilde{\mathbf{y}}$, and hence $g(\mathbf{x}; \widetilde{\mathbf{y}})$ is $L_g$-smooth.

   - For the upper-level function $f(\mathbf{x}; \widetilde{\mathbf{y}})$, note that $\nabla^2_{\widetilde{\mathbf{y}}} f(\mathbf{x}; \widetilde{\mathbf{y}}) = \frac{L_f}{L} M$, where $M \in \mathbb{R}^{n(T+1) \times n(T+1)}$ is a tri-diagonal matrix, where the absolute value of each nonzero element is bounded by some numerical constant, due to the fact that $C_r$ and $C_l$ are numerical constants, and that the functions $\Phi$ and $\Psi$, together with their derivatives, are bounded by numerical constants, as shown in item 3 of Lemma 1. Then, we have $\|M\|_2 \leq C_M$ for some numerical constant $C_M > 0$. Thus, choosing $L = C_M$ yields $\|\nabla^2_{\widetilde{\mathbf{y}}} f(\mathbf{x}; \widetilde{\mathbf{y}})\|_2 \leq L_f$. Since $f(\mathbf{x}; \widetilde{\mathbf{y}})$ depends only on $\widetilde{\mathbf{y}}$, it is thus $L_f$-smooth.

3. Next, we need to show that the gradient norm $\|\nabla_{\widetilde{\mathbf{y}}} f(\mathbf{x}; \widetilde{\mathbf{y}})\|_2$ is bounded by a numerical constant that is independent of both $T$ and $n$. This step is particularly challenging. For example, the previous lower bound in Ji & Liang 2023 circumvents this requirement by exploiting the strong convexity of the hyper-objective to guarantee gradient boundedness during the optimization process. However, that strategy applies only to the strongly-convex–strongly-convex setting and may not extend well to nonconvex or stochastic regimes. Moreover, another lower bound in Kwon et al. 2024 sets the upper-level function as a scalar $y$, which ensures that the gradient norm remains bounded by a constant.

For our construction in Equation (5), it can be obtained that $\|\nabla_{\widetilde{\mathbf{y}}} f(\mathbf{x}; \widetilde{\mathbf{y}})\|_2 = \frac{\lambda L_f}{L} \mathbf{v}$, where $\mathbf{v} \in \mathbb{R}^{n(T+1)}$ has at most $2(T+1)$ nonzero entries at coordinates $kn+1$ for $k = 0, \ldots, T$ and $jn$ for $j = 1, \ldots, T+1$.

Moreover, the absolute value of each nonzero entry is bounded by a positive numerical constant, owing to the fact that $C_r$ and $C_l$ are numerical constants and that $\Psi, \Psi', \Phi,$ and $\Phi'$ are all bounded (Lemma 1, item 3). Therefore, we have $\|v\| \leq C_0 \sqrt{T}$ for some numerical constant $C_0$. Thus, we have $\|\nabla_{\widetilde{\mathbf{y}}} f(\mathbf{x}; \widetilde{\mathbf{y}})\|_2 \leq \frac{C_0 L_f}{L} \lambda \sqrt{T}$. As will be seen later, $T$ is chosen such that $\lambda \sqrt{T} \leq \sqrt{\frac{\Delta L}{12 L_f}}$, which, together with $\frac{\Delta}{L_f} = \mathcal{O}(1)$, implies that $\|\nabla_{\widetilde{\mathbf{y}}} f(\mathbf{x}; \widetilde{\mathbf{y}})\|_2 = \mathcal{O}(1)$.

**Zero-chain properties and iterate subspaces.** We initialize $\mathbf{x}$ and $\widetilde{\mathbf{y}}$ to be $\mathbf{0}$. Then, based on the tri-diagonal structure of $A$ and the properties of $\Psi$ function in Lemma 1 (item 1), it can be quickly verified from our construction in Equation (5) that

- At the first iteration, $y_n^{(0)}$ becomes activated, because $x_0 \neq 0$ and $\partial g / \partial y_n^{(0)} = -L_g x_0$. Thus, at the second iteration, $y_1^{(1)}$ becomes activated due to the zero-chain property of the $f(\mathbf{x}; \widetilde{\mathbf{y}})$ function.

- Suppose the iterates have begun updating $\mathbf{y}^{(i)}$ but have not yet reached $y_n^{(i)}$ (i.e., $y_n^{(i)} = 0$) for some $i \geq 1$. This implies that $y_n^{(j)} = 0$ for all $j \geq i$. Then, by Lemma 1 (item 1), it is verified that for all $j \geq i$,

$$\frac{\partial f(\mathbf{x}; \widetilde{\mathbf{y}})}{\partial y_1^{(j+1)}} = -\frac{C_r \lambda L_f}{L} \Psi\left(-\frac{C_l}{\lambda} y_n^{(j)}\right) \Phi'\left(-\frac{C_r}{\lambda} y_1^{(j+1)}\right)$$
$$- \frac{C_r \lambda L_f}{L} \Psi\left(\frac{C_l}{\lambda} y_n^{(j)}\right) \Phi'\left(\frac{C_r}{\lambda} y_1^{(j+1)}\right) = 0,$$

which, together with the structure of the lower-level function and the condition $x_j = 0$ for all $j \geq i$, implies that $\mathbf{y}^{(j)} = \mathbf{0}$ for all $j \geq i+1$. This property is crucial because it preserves the zero-chain structure along the sequence $\{\mathbf{y}^{(i)}\}_{i=1}^T$ and ensures that advancing from one adjacent $\mathbf{y}$-iterate to the next necessarily requires at least $n$ iterations.

- Suppose the iterates have begun updating $\mathbf{y}^{(i)}$ but have not yet reached $y_n^{(i)}$ (i.e., $y_n^{(i)} = 0$) for some $i \geq 1$. Then, for all $j \geq i$, the gradient of $g(\mathbf{x}; \widetilde{\mathbf{y}})$ with respect to $x_j$ is given by $-y_n^{(j)}$. As a consequence, the coordinate $x_j$ won't be activated until $y_n^{(j)}$ is activated.

Based on the above analysis, it can be derived that at any iteration $Kn + k$ with $K = 0, ..., T-1$ and $k = 1, ..., n$,

$$\text{supp}(\mathbf{y}^{(i)}) \subseteq \{1, ..., n\}, \quad i \le K \text{ and } i \ne 0$$
$$\text{supp}(\mathbf{y}^{(K+1)}) \subset \{1, ..., k\}$$
$$\text{supp}(\mathbf{y}^{(i)}) = \emptyset, \quad i > K+1$$
$$\text{supp}(\mathbf{x}) \subset \{0, ..., K\}. \tag{6}$$

Accordingly, to activate all coordinates of $\mathbf{x}$, one must perform at least $Tn$ iterations in total.

**The hyper-objective function and its key properties.** First, we can verify that the lower-level solutions are given by

$$(\mathbf{y}^{(i)})^* = \underbrace{\frac{4n^2 + 1}{n^2}\left(\frac{1}{n^2}I_n + A\right)^{-1}}_{M}\mathbf{b}_x^{(i)}.$$

The hyper-objective $H(x) := f(x; \widetilde{\mathbf{y}}^*)$ is then given by

$$H(\mathbf{x}) = \sum_{i=1}^{T}\frac{\lambda^2 L_f}{L}\Big[\Psi\big(-\frac{C_l}{\lambda}M_{n,n}x_{i-1}\big)\Phi\big(-\frac{C_r}{\lambda}M_{1,n}x_i\big)$$
$$- \Psi\big(\frac{C_l}{\lambda}M_{n,n}x_{i-1}\big)\Phi\big(\frac{C_r}{\lambda}M_{1,n}x_i\big)\Big].$$

Note that the above definition of $H(\mathbf{x})$ involves the quantities $M_{n,n}$ and $M_{1,n}$, whose behaviors are characterized in the following lemma.

**Lemma 2.** *Let $A \in \mathbb{R}^{n \times n}$ be the tri-diagonal matrix defined by Equation* (3) *and define $S := \left(A + \frac{1}{n^2}I_n\right)^{-1}$. Then for every integer $n \ge 1$,*

$$c\,n \le S_{1,n}, S_{n,n} \le C\,n,$$
$$c := 1 - \frac{\pi^2}{12}, \; C := 1 + \frac{\pi^2}{12}. \tag{7}$$

Based on Lemma 2 and $4 \le \frac{4n^2+1}{n^2} \le 5$, it can be derived that $4cn \le M_{1,n}, M_{n,n} \le 5Cn$, where $c$ and $C$ are given by Equation (7). Thus, choose numerical constants $C_l$ and $C_r$ such that

$$\frac{C_l M_{n,n}}{n} = \frac{C_r M_{1,n}}{n} = \widetilde{C}, \tag{8}$$

where $\widetilde{C}$ is a numerical constant. Then, we use the following lemma to provide a lower bound on the gradient norm when the algorithm has not yet reached the end of the chain.

**Lemma 3.** *If $|x_i| < \frac{\lambda}{\widetilde{C}n}$ for some $i \le T$. Then, we have $\|\nabla H(\mathbf{x})\|_2 \ge \frac{\lambda L_f \widetilde{C}n}{L}$.*

The following lemma provides the bound on the optimality gap of the hyper-objective function $H(\mathbf{x})$:

**Lemma 4.** *The hyper-objective function $H(\mathbf{x})$ satisfies $H(\mathbf{0}) - \inf_{\mathbf{x}} H(\mathbf{x}) \le \frac{12\lambda^2 L_f T}{L}$.*

Based on all the above auxiliary lemmas, we begin to prove our main theorem.

**Proof of Theorem 4.1.** First note that if $x_T = 0$, based on Lemma 3, we have that $\|\nabla H(\mathbf{x})\|_2 \ge \frac{\lambda L_f \widetilde{C}n}{L}$. Choosing $\lambda = \frac{\epsilon L}{L_f \widetilde{C}n}$ guarantees $\|\nabla H(\mathbf{x})\|_2 \ge \epsilon$. Then, we need to verify that $H(\mathbf{0}) - \inf_{\mathbf{x}} H(\mathbf{x}) \le \Delta$. Based on Lemma 4, we have that $H(\mathbf{0}) - \inf_{\mathbf{x}} H(\mathbf{x}) \le \frac{12\lambda^2 L_f T}{L}$, which, by setting $T = \left\lfloor \frac{\Delta L}{12\lambda^2 L_f} \right\rfloor$, guarantees that $H(\mathbf{0}) - \inf_{\mathbf{x}} H(\mathbf{x}) \le \Delta$.

Based on the subspace analysis in Equation (6), we have that $x_T = 0$ if $t < Tn$, and hence $\|\nabla H(\mathbf{x}^t)\|_2 \ge \epsilon$. Recall that $n = \left\lfloor \sqrt{\frac{L_g - \mu}{4\mu}} \right\rfloor$. Thus, to achieve an $\epsilon$-accurate stationary solution, there are at least

$$Tn = \frac{c_0 \Delta n^3}{\epsilon^2} = \frac{\Delta L n}{12 L_f}\frac{L_f^2 \widetilde{C}^2 n^2}{\epsilon^2 L^2} = \frac{C_0 \Delta L_f \kappa^{\frac{3}{2}}}{\epsilon^2}$$

oracle calls, where $c_0$ is some numerical constant. Then, the proof is complete. $\square$

# 5. Lower Bounds in Stochastic Setting

In this section, we provide a lower bound for stochastic first-order oracles. We first introduce several important definitions and lemmas from Arjevani et al. 2023, serving as the foundation for our constructions in the stochastic setting.

## 5.1. Auxiliary Definitions and Lemmas

Following Arjevani et al. 2023, to establish a lower bound in the stochastic setting, we adopt the notion of a probability-$p$ zero-chain.

**Definition 6** (Probability-$p$ zero-chain). *A function $f : \mathcal{X} \to \mathbb{R}$ with a stochastic first-order oracle $O : \mathbf{x} \mapsto (f(\mathbf{x}), G_f(\mathbf{x}; \xi))$ is a probability-$p$ zero-chain if*

$$\text{supp}(\mathbf{x}) \subset \{1, \ldots, i-1\}$$
$$\implies \begin{cases} \mathbb{P}\big(\text{supp}(G_f(\mathbf{x}; \xi)) \not\subset \{1, \ldots, i-1\}\big) \le p, \\ \mathbb{P}\big(\text{supp}(G_f(\mathbf{x}; \xi)) \subset \{1, \ldots, i\}\big) = 1. \end{cases}$$

The above definition implies that at each iteration, a new coordinate $i$ becomes activated (i.e., the iterate acquires a nonzero entry at coordinate $i$) with probability $p$. The following lemma (which is an adapted version from Li et al. 2021) provides a recipe for constructing a probability-$p$ zero-chain based on a given zero-chain.

**Lemma 5** (Lemma 3 in Arjevani et al. 2023). *Let $f : \mathcal{X} \to \mathbb{R}$ be a zero-chain on $\mathcal{X} \subset \mathbb{R}^T$. For $\mathbf{x} \in \mathcal{X}$, let $i^*(\mathbf{x}) := \inf\{i \in [T] : x_i = 0\}$ be the next coordinate to activate. For $p \in (0, 1]$, define the stochastic gradient estimator*

$G_f(\mathbf{x}; \xi)$ *coordinate-wisely by*

$$[G_f(\mathbf{x}, \xi)]_i := \begin{cases} \dfrac{\xi}{p} \nabla_i f(\mathbf{x}), & \textit{if } i = i^*(\mathbf{x}), \\ \nabla_i f(\mathbf{x}), & \textit{otherwise,} \end{cases}$$

*where $\xi \sim \mathrm{Bernoulli}(p)$. Suppose there exists $G < \infty$ such that $\|\nabla f(\mathbf{x})\|_\infty \le G$ for all $\mathbf{x} \in \mathcal{X}$. Then, the oracle $O : \mathbf{x} \mapsto (f(\mathbf{x}), G_f(\mathbf{x}, \xi))$ is a stochastic first-order oracle with bounded variance $\sigma^2 \le G^2(1-p)/p$. Moreover, $f$ with oracle $O$ is a probability-$p$ zero-chain.*

Lemma 5 allows us to build a probability-$p$ zero-chain based on the zero-chain we establish in Equation (5) and Figure 1. However, as also noted by Li et al. 2021 for min-max problems, one main challenge lies in the unboundedness of the iterates $\mathbf{x}$ and $\widetilde{\mathbf{y}}$, such that the gradient norm of the lower-level function $\|\nabla g(\mathbf{x}; \widetilde{\mathbf{y}})\|_\infty$ is unbounded. To this end, Li et al. 2021 modify the quadratic components in their deterministic worst-case instance and introduce two bounded hypercubes as the domains for $\mathbf{x}, \mathbf{y}$: $\mathcal{C}_{R_x}^m := \{\mathbf{x} \in \mathbb{R}^m : \|\mathbf{x}\|_\infty \le R_x\}$ and $\mathcal{C}_{R_y}^n := \{\mathbf{y} \in \mathbb{R}^n : \|\mathbf{y}\|_\infty \le R_y\}$, where $R_x$ and $R_y$ are chosen so that the variance of the stochastic oracle is bounded by $G$. Interestingly, unlike Li et al. 2021, which must revise the quadratic components in their deterministic construction, our deterministic instance in Equation (5) can be used directly, provided that the domain radius $R_x$ and $R_y$ are properly selected, as seen in our analysis later.

### 5.2. Main Result: A Lower Bound on Stochastic First-Order Oracle Complexity

The following theorem establishes a complexity lower bound for stochastic first-order bilevel algorithms.

**Theorem 5.1.** *For any $L_f, L_g, \mu, \Delta, \epsilon > 0$ satisfying $\kappa = L_g/\mu \ge 1$ and $\frac{\Delta}{L_f} = \mathcal{O}(1)$, there exist functions $f : \mathcal{X} \times \mathcal{Y} \to \mathbb{R}$ and $g : \mathcal{X} \times \mathcal{Y} \to \mathbb{R}$ such that $\{f, g\} \in \mathcal{F}(L_f, L_g, \mu, \Delta)$ for some $\mathcal{X} \subset \mathbb{R}^m$ and $\mathcal{Y} \subset \mathbb{R}^n$, and stochastic first-order oracles $O$ for both $f$ and $g$ such that for any first-order bilevel algorithm of the form in Definition 4, in order to find an $\epsilon$-accurate stationary point $\mathbf{x}$ such that $\mathbb{E}\left[L_h\|\mathcal{P}_\mathcal{X}[\mathbf{x} - (1/L_h)\nabla H(\mathbf{x})] - \mathbf{x}\|_2\right] < \epsilon$, the algorithm must use at least*

$$\Omega\left(\frac{L_f^3 \Delta \kappa^{5/2} \sigma^2}{L_g^2 \epsilon^4}\right) \tag{9}$$

*stochastic oracle calls, where $L_h$ is the smoothness parameter of the hyper-objective $H(\mathbf{x})$.*

The proof outline is provided in Section B. In the stochastic setting, Arjevani et al. 2023 establish a lower bound of $\Omega(1/\epsilon^4)$ for smooth nonconvex optimization, and Li et al. 2021 prove a lower bound of $\Omega(\kappa^{1/3}/\epsilon^4)$ for smooth nonconvex-strongly-concave min-max optimization. For

smooth nonconvex-strongly-convex bilevel optimization, we obtain in Theorem 5.1 a significantly larger lower bound of $\Omega(\kappa^{5/2}/\epsilon^4)$. To the best of our knowledge, this is the first lower-bound result for stochastic bilevel optimization showing that the nonconvex-strongly-convex bilevel optimization is strictly more challenging than both smooth nonconvex optimization and smooth nonconvex-strongly-concave min-max optimization in the stochastic setting.

In what follows, we discuss connections and comparisons with existing lower bounds.

(1) Kwon et al. (2024) establish a lower bound of $\Omega(\epsilon^{-6})$ for bilevel optimization under a so-called $\mathbf{y}^*$-aware stochastic first order oracle with bounded variance. Their hard instance is constructed as

$$f(\mathbf{x}; y) = y, \ g(\mathbf{x}; y) = (y - F(\mathbf{x}))^2, \ \mathbf{x} \in \mathbb{R}^{\epsilon^{-2}}, y \in \mathbb{R}$$

where function $F(\mathbf{x}) = \epsilon^2 \sum_{i=1}^{\epsilon^{-2}} \left[\Psi(-x_{i-1})\Phi(-x_i) - \Psi(x_{i-1})\Phi(x_i)\right]$. It can be verified that $|F(\mathbf{x})| = \mathcal{O}(1)$, and therefore

$$\|\nabla_{\mathbf{x}, y}^2 g(\mathbf{x}; y)\|_2 = \mathcal{O}\left(\epsilon^2 \sqrt{\epsilon^{-2}}\right) = \mathcal{O}(\epsilon).$$

In addition, their $\mathbf{y}^*$-aware oracle requires $\|y - y^*\| = \mathcal{O}(\epsilon)$, such that $|g(\mathbf{x}; y)|$ is of order $\mathcal{O}(\epsilon)$. These conditions can be approximately satisfied in our construction by choosing $L_g = \mathcal{O}(\epsilon)$, since $\|\nabla_{\mathbf{x}, \widetilde{\mathbf{y}}}^2 g(\mathbf{x}; \widetilde{\mathbf{y}})\|_2 = L_g$ and both $\mathbf{x}$ and $\widetilde{\mathbf{y}}$ are bounded. By this choice, our Theorem 5.1 also yields a lower bound of order $\Omega(\epsilon^{-6})$.

In contrast, a more standard and practically relevant setting assumes $L_g, L_f = \Theta(1)$, independent of $\epsilon$ or the condition number $\kappa$. Under this commonly studied regime, obtaining an $\Omega(\epsilon^{-6})$ lower bound for bilevel optimization remains an open problem.

## 6. Conclusion

In this work, we developed new hard instances that establish improved lower bounds for smooth nonconvex and strongly convex bilevel optimization under both deterministic and stochastic first order oracle models. Our results demonstrate that bilevel optimization is fundamentally more challenging than classical single-level and min-max formulations, and they reveal significant separations between the best known upper and lower bounds. These findings highlight that the current theoretical understanding of bilevel optimization is still far from complete.

## Impact Statement

his paper presents work whose goal is to advance the field of bilevel optimization theory. There are many potential societal consequences of our work, none which we feel must be specifically highlighted here.

## Acknowledgments

K. Ji was partially supported by NSF grants CCF-2311274 and ECCS-2326592.

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

# Appendix

## A. Some Discussions on Future Works

There are several promising directions for future research. First, even for the simplified and practically meaningful setting in which the lower level function is quadratic, the optimal complexity remains open. We suggest that closing these gaps may require first studying this simpler yet meaningful quadratic setting. Moreover, our constructions suggest that sharper lower and upper bounds may be obtained by designing algorithms that exploit higher-order structure of the lower-level function. Second, closing the large gaps between the existing upper bounds and our lower bounds, especially the gap of order $\kappa^2$ in the deterministic case and the dependence on $\epsilon$ in the stochastic case, represents an important challenge. Third, another compelling direction is to investigate whether an $\Omega(\epsilon^{-6})$ lower bound can be achieved under the standard regime where the smoothness constants $L_f$ and $L_g$ are $\Theta(1)$, a question that remains unresolved. Finally, extending the lower bound framework to broader variants of bilevel optimization, including settings with constraints, approximate inner solvers, or distributed architectures, may deepen the understanding of the fundamental limits of bilevel learning.

Overall, we hope that the insights developed in this paper serve as a starting point for further studies toward a complete theory of the computational complexity of bilevel optimization.

## B. Analysis and Proof Outline for Stochastic Lower Bound

We use the following construction $\{f_{sc}(\mathbf{x}; \widetilde{\mathbf{y}}), g_{sc}(\mathbf{x}; \widetilde{\mathbf{y}})\}$ as the hard instance in the stochastic setting. For any $\mathbf{x} \in \mathcal{C}^T_{r_x \lambda / n}$ and $\widetilde{\mathbf{y}} \in \mathcal{C}^{n(T+1)}_{r_y \lambda}$,

$$f_{sc}(\mathbf{x}; \widetilde{\mathbf{y}}) = \sum_{i=1}^{T} \frac{\lambda^2 L_f}{L} \left[ \Psi\left(-\frac{C_l}{\lambda} y_n^{(i-1)}\right) \Phi\left(-\frac{C_r}{\lambda} y_1^{(i)}\right) - \Psi\left(\frac{C_l}{\lambda} y_n^{(i-1)}\right) \Phi\left(\frac{C_r}{\lambda} y_1^{(i)}\right) \right]$$

$$g_{sc}(\mathbf{x}; \widetilde{\mathbf{y}}) = \sum_{i=0}^{T} \left[ \frac{L_g n^2}{2(4n^2+1)} (\mathbf{y}^{(i)})^\top \left(\frac{1}{n^2} I_n + A\right) \mathbf{y}^{(i)} - L_g (\mathbf{b}_x^{(i)})^\top \mathbf{y}^{(i)} \right], \tag{10}$$

where $r_x$ and $r_y$ are positive numerical constants from the hypercube sizes, chosen such that $r_y \geq 10 r_x$ and $r_x > \frac{1}{\widetilde{C}}$, where $\widetilde{C} > 0$ is the numerical constant defined in Equation (8). The constants $C_l, C_r,$ and $L$ are the same as in the deterministic setting. The parameter $\lambda$ is selected to satisfy $\lambda \sqrt{T} = \mathcal{O}(1)$, and its exact form will be specified later. Recall that $x_0 = \frac{\lambda}{\widetilde{C} n} < \frac{r_x \lambda}{n} \in \mathcal{C}^1_{r_x \lambda / n}$.

The following lemma shows that, with appropriately chosen $r_x$ and $r_y$, the lower-level minimizer $\widetilde{\mathbf{y}}^*$ lies within the selected bounded domain.

**Lemma 6.** *If $r_y \geq 10 r_x$, the lower-level minimizer $\widetilde{\mathbf{y}}^*$ of the instance in Equation (10) satisfies $\widetilde{\mathbf{y}}^* \in \mathcal{C}^{n(T+1)}_{r_y \lambda}$.*

Building on Lemma 6, we establish the following lemma, which provides several properties of the instance in Equation (10) that will be used in the proof of the main theorem.

**Lemma 7.** *Suppose $r_y \geq 10 r_x$, $r_x > \frac{1}{\widetilde{C}}$, and $\lambda \sqrt{T} = \mathcal{O}(1)$. The functions $f_{sc}$ and $g_{sc}$ satisfy:*

*(a) $f_{sc}$ and $g_{sc}$ satisfy all items 1-4 in Definition 1.*

*(b) $H_{sc}(\mathbf{0}) - \min_\mathbf{x} H_{sc}(\mathbf{x}) \leq \frac{12 \lambda^2 L_f T}{L}$.*

*(c) $H_{sc}(\mathbf{x})$ is $L_h$-smooth with $L_h = \frac{c_0 n^2 L_f}{L}$ for some numerical constant $c_0$.*

*(d) For any $(\mathbf{x}, \widetilde{\mathbf{y}}) \in \mathcal{C}^T_{r_x \lambda / n}, \times \mathcal{C}^{n(T+1)}_{r_y \lambda}$, we have $\|\nabla_{\widetilde{\mathbf{y}}} f_{sc}(\mathbf{x}; \widetilde{\mathbf{y}})\|_\infty \leq \frac{c_1 \lambda L_f}{L}, \|\nabla_{\widetilde{\mathbf{y}}} g_{sc}(\mathbf{x}; \widetilde{\mathbf{y}})\|_\infty \leq 2 L_g r_y \lambda$, and $\|\nabla_\mathbf{x} g_{sc}(\mathbf{x}; \widetilde{\mathbf{y}})\|_\infty \leq L_g r_y \lambda$, where $c_1$ is a numerical constant.*

Similarly to Lemma 3, we then provide a lower bound of the hyper-gradient norm when the algorithm has not yet reached the end of the chain.

**Lemma 8.** *Suppose $r_x > \frac{1}{\tilde{C}}$. If $x_i < \frac{\lambda}{\tilde{C}n}$ for some $i \leq T$, then, we have*

$$L_h \|\mathcal{P}_{\mathcal{X}}[\mathbf{x} - (1/L_h)\nabla H_{sc}(\mathbf{x})] - \mathbf{x}\|_2 \geq \frac{c_2 L_f n\lambda}{L},$$

*where $c_2 > 0$ is a numerical constant.*

Based on all the above auxiliary lemmas, we begin to prove our main theorem.

**Proof of Theorem 5.1.** Based on part (d) of Lemma 7, we now construct a probability-$p$ zero-chain following the approach of Arjevani et al. 2023, with a slight modification. In Arjevani et al. 2023, the key idea is to perturb the gradient only at the next coordinate to be activated, so that this coordinate is revealed with probability $p$. For our zero-chain given in Equation (6), let $i^* \in \{n+1, ..., (T+1)n\}$ be the next coordinate to activate. Thus, we can define the stochastic gradient as follows.

- When $i^* \bmod n \neq 1$, perturb the gradients at the coordinate $i = i^*$ as $\frac{\xi}{p} \frac{\partial g_{sc}(\mathbf{x}; \widetilde{\mathbf{y}})}{\partial \widetilde{y}_i}$, where $\xi \sim \text{Bernoulli}(p)$. The gradients at all other coordinates remain unchanged and receive no perturbation.

- When $i^* \bmod n = 1$, perturb the gradients at the coordinate $i = i^*$ as $\frac{\xi}{p} \frac{\partial g_{sc}(\mathbf{x}; \widetilde{\mathbf{y}})}{x_j}$, where $j = (i^* - 1)/n$ and $\xi \sim \text{Bernoulli}(p)$. The gradients at all other coordinates remain unchanged and receive no perturbation.

Note that in the above stochastic oracles, we **do not** perturb the gradients of $f$. It can be verified that the stochastic gradients defined above are unbiased. Using Lemma 5 together with part (d) of Lemma 7, we conclude that our construction in Equation (10), equipped with these stochastic oracles, forms a probability-$p$ zero-chain, and the variance of the oracles is bounded by

$$c_3 L_g^2 \lambda^2 \left(\frac{1-p}{p}\right),$$

where the bound follows from $(d)$ of Lemma 7, and $c_3$ is a positive numerical constant. Thus, to ensure the variance is bounded by $\sigma^2$, it suffices to choose

$$p = \min\left\{1, c_3 \frac{L_g^2 \lambda^2}{\sigma^2}\right\}. \tag{11}$$

Then, based on Lemma 9 and the stochastic oracles constructed above, we have that with probability $1 - \delta$, $x_T = 0$ if

$$t \leq \frac{(n-1)T - 1 - \log(\frac{1}{\delta})}{2p}. \tag{12}$$

Based on the choice of $p$ in Equation (11), we have

$$\frac{(n-1)T - 1 - \log(\frac{1}{\delta})}{2p} \geq \frac{((n-1)T - 1 - \log(\frac{1}{\delta}))\sigma^2}{2c_3 L_g^2 \lambda^2},$$

which, together with Equation (12), yields that with probability $1 - \delta$, $x_T = 0$ for all

$$t \leq \frac{((n-1)T - 1 - \log(\frac{1}{\delta}))\sigma^2}{2c_3 L_g^2 \lambda^2}.$$

This, with Lemma 8, implies that with probability $1 - \delta$, $x_T = 0$ for all $t \leq \frac{((n-1)T - 1 - \log(\frac{1}{\delta}))\sigma^2}{2c_3 L_g^2 \lambda^2}$, and hence

$$L_h \|\mathcal{P}_{\mathcal{X}}[\mathbf{x}^t - (1/L_h)\nabla H_{sc}(\mathbf{x}^t)] - \mathbf{x}^t\|_2 \geq \frac{c_2 L_f n\lambda}{L},$$

which, by setting $\lambda = \frac{2L\epsilon}{c_2 L_f n}$, yields that $L_h\|\mathcal{P}_\mathcal{X}[\mathbf{x}^t - (1/L_h)\nabla H_{sc}(\mathbf{x}^t)] - \mathbf{x}^t\|_2 \geq 2\epsilon$. Set $\delta = \frac{1}{2}$. Then, for all $t \leq \frac{((n-1)T - 1 - \log(\frac{1}{\delta}))\sigma^2}{2c_3 L_g^2 \lambda^2}$,

$$\mathbb{E}\left[L_h\|\mathcal{P}_\mathcal{X}[\mathbf{x}^t - (1/L_h)\nabla H_{sc}(\mathbf{x}^t)] - \mathbf{x}^t\|_2\right] \geq \frac{1}{2}(2\epsilon) = \epsilon.$$

Based on $(b)$ of Lemma 7, we have $\frac{12\lambda^2 L_f T}{L} = \Delta$, which implies that $T = \frac{\Delta L}{12\lambda^2 L_f}$. Thus, to achieve an $\epsilon$-accurate stationary point, the algorithm must use at least

$$\Omega\left(\frac{nT\sigma^2}{L_g^2\lambda^2}\right) = \Omega\left(\frac{n\Delta\sigma^2}{L_f L_g^2\lambda^4}\right) = \Omega\left(\frac{n^5 L_f^3 \Delta\sigma^2}{L_g^2\epsilon^4}\right),$$

which, together with $n = \sqrt{\kappa}$, finishes the proof. $\qquad\square$

## A. Proofs for Deterministic Lower Bound

### A.1. Proof of Lemma 2

It is straightforward to verify that $c \leq S_{1,1} = 1 \leq C$, so the claim holds for $n = 1$. For the remainder of the proof, we assume $n \geq 2$. Set $s := 1/n^2$. The eigenpairs of $A$ are

$$\mu_k = 2\left(1 - \cos\left(\frac{(k-1)\pi}{n}\right)\right), \qquad k = 1, \ldots, n,$$

with orthonormal eigenvectors

$$q_1(j) = \frac{1}{\sqrt{n}}, \qquad q_k(j) = \sqrt{\frac{2}{n}} \cos\left(\frac{(k-1)(j-\frac{1}{2})\pi}{n}\right), \ k \geq 2.$$

Thus

$$A = Q\Lambda Q^\top, \qquad \Lambda = \mathrm{diag}(\mu_1, \ldots, \mu_n), \qquad Q = [q_1 \ \ldots \ q_n].$$

Hence

$$S = (A + sI_n)^{-1} = Q(\Lambda + sI_n)^{-1}Q^\top = \sum_{k=1}^n \frac{1}{\mu_k + s} q_k q_k^\top,$$

so we can express $S_{i,j}$ as

$$S_{i,j} = \sum_{k=1}^n \frac{q_k(i)q_k(j)}{\mu_k + s}.$$

Note that $\frac{q_1(i)q_1(j)}{s} = \frac{1/n}{1/n^2} = n$. Thus, we have

$$S_{i,j} = n + R_{i,j}, \qquad R_{i,j} := \sum_{k=2}^n \frac{q_k(i)q_k(j)}{\mu_k + s}.$$

**Higher eigenmodes.** Because $|q_k(\cdot)| \leq \sqrt{2/n}$,

$$|q_k(i)q_k(j)| \leq \frac{2}{n}.$$

Also for $k \geq 2$,

$$\mu_k = 2(1 - \cos(\frac{(k-1)\pi}{n})) \geq \frac{4(k-1)^2}{n^2},$$

so

$$\frac{1}{\mu_k + s} \leq \frac{n^2}{4(k-1)^2}.$$

Thus

$$|R_{i,j}| \leq \sum_{k=2}^n \frac{2}{n} \cdot \frac{n^2}{4(k-1)^2} = \frac{n}{2}\sum_{m=1}^{n-1}\frac{1}{m^2} \leq \frac{\pi^2}{12}n.$$

**Final bounds.**

$$S_{n,n} = n + R_{n,n}, \ R_{n,n} \geq 0, \qquad S_{1,n} = n + R_{1,n}, \ |R_{1,n}| \leq \frac{\pi^2}{12}n.$$

Hence for all $n \geq 2$,

$$\left(1 - \frac{\pi^2}{12}\right)n \leq S_{1,n}, S_{n,n} \leq \left(1 + \frac{\pi^2}{12}\right)n.$$

Then, the proof is complete.

### A.2. Proof of Lemma 3

Note that $x_0 = \frac{\lambda}{C_l M_{n,n}} = \frac{\lambda}{\widetilde{C}n}$. Since $|x_0| \geq \frac{\lambda}{\widetilde{C}n}$ and $|x_i| < \frac{\lambda}{\widetilde{C}n}$, we can find some $0 < j \leq i$ such that $|x_{j-1}| \geq \frac{\lambda}{\widetilde{C}n}$ and $|x_j| < \frac{\lambda}{\widetilde{C}n}$. Thus, look at

$$\frac{\partial H(\mathbf{x})}{\partial x_j} = -\frac{\lambda L_f \widetilde{C}n}{L} \left[\Psi\left(-\frac{\widetilde{C}n}{\lambda}x_{j-1}\right)\Phi'\left(-\frac{\widetilde{C}n}{\lambda}x_j\right) + \Psi\left(\frac{\widetilde{C}n}{\lambda}x_{j-1}\right)\Phi'\left(\frac{\widetilde{C}n}{\lambda}x_j\right)\right]$$
$$-\frac{\lambda L_f \widetilde{C}n}{L} \left[\Psi'\left(-\frac{\widetilde{C}n}{\lambda}x_j\right)\Phi\left(-\frac{\widetilde{C}n}{\lambda}x_{j+1}\right) + \Psi'\left(\frac{\widetilde{C}n}{\lambda}x_j\right)\Phi\left(\frac{\widetilde{C}n}{\lambda}x_{j+1}\right)\right],$$

which, in conjunction with Lemma 1 (items 2 and 4), implies that

$$\|\nabla H(\mathbf{x})\|_2 \geq \left|\frac{\partial H(\mathbf{x})}{\partial x_j}\right| \geq \frac{\lambda L_f \widetilde{C}n}{L}.$$

Then, the proof is complete.

### A.3. Proof of Lemma 4

First note that

$$H(\mathbf{0}) = \frac{\lambda^2 L_f}{L} \left[\left(\Psi\left(-\frac{C_l}{\lambda}M_{n,n}x_0\right) - \Psi\left(\frac{C_l}{\lambda}M_{n,n}x_0\right)\right)\Phi(0)\right] \leq 0, \tag{13}$$

where the inequality follows because $\frac{C_l}{\lambda}M_{n,n}x_0 \geq 0$ and from the definitions of $\Psi$ and $\Psi$ functions in Equation (4). Furthermore, based on Lemma 1 (item 4), we have that

$$H(\mathbf{x}) \geq -\frac{\lambda^2 L_f}{L} \sum_{i=1}^{T} \Psi\left(\frac{C_l}{\lambda}M_{n,n}x_{i-1}\right)\Phi\left(\frac{C_r}{\lambda}M_{1,n}x_i\right) \geq -\frac{12\lambda^2 L_f T}{L}, \tag{14}$$

which, combined with $H(\mathbf{0}) \leq 0$, implies that

$$H(\mathbf{0}) - \inf_{\mathbf{x}} H(\mathbf{x}) \leq \frac{12\lambda^2 L_f T}{L},$$

which finishes the proof.

## B. Proofs for Stochastic Lower Bound

### B.1. Auxiliary Lemmas

For a probability-$p$ zero-chain, at each iteration, a new coordinate is discovered with probability at most $p$. Therefore, it takes at least $1/p$ steps in expectation to activate a new coordinate. The following lemma, adapted from Arjevani et al. 2023; Li et al. 2021, shows that at least $\Omega(T/p)$ iterations are required to reach the end of a probability-$p$ zero-chain.

**Lemma 9** (Lemma 1 in Arjevani et al. 2023). *Let $f : \mathcal{X} \to \mathbb{R}$, where $\mathcal{X} \subset \mathbb{R}^T$ satisfies $\text{supp}\left(P_{\mathcal{X}}(\mathbf{x})\right) = \text{supp}(\mathbf{x}), \ \forall \mathbf{x} \in \mathbb{R}^T$, and suppose $f$ is a probability-$p$ zero-chain with a stochastic first-order oracle. Then, for any first-order algorithm, with probability at least $1 - \delta$, the $T$-th coordinate of $\mathbf{x}$ at the $t^{th}$ iteration, satisfies*

$$x_T^t = 0, \qquad \forall t \leq \frac{T - \log(1/\delta)}{2p}.$$

**Lemma 10.** *Recall* $S := \left(A + \frac{1}{n^2} I_n\right)^{-1}$. *For every* $i = 1, \ldots, n$,

$$S_{1,n} \leq S_{i,n} \leq S_{n,n}.$$

*Proof.* Define $B := A + \frac{1}{n^2} I_n$, and $\mathbf{v} \in \mathbb{R}^n$ the last column of $S$, i.e., $\mathbf{v} := S_{\cdot,n}$, $v_i := S_{i,n}$, $i = 1, \ldots, n$. Since $\mathbf{v}$ is the last column of $S = B^{-1}$, it solves the linear system

$$B\mathbf{v} = \mathbf{e}_n.$$

Writing this componentwise, we obtain

$$\begin{aligned}
(1 + n^{-2})v_1 - v_2 &= 0, \\
-v_{i-1} + (2 + n^{-2})v_i - v_{i+1} &= 0, \qquad i = 2, \ldots, n-1, \\
-v_{n-1} + (1 + n^{-2})v_n &= 1.
\end{aligned}$$

Define the forward differences

$$d_i := v_{i+1} - v_i, \qquad i = 1, \ldots, n-1.$$

We next derive a system of equations for $\mathbf{d} = (d_1, \ldots, d_{n-1})^\top$. For $i = 2, \ldots, n-2$, subtracting the equation at index $i$ from that at index $i + 1$ gives

$$\left(-v_i + (2 + n^{-2})v_{i+1} - v_{i+2}\right) - \left(-v_{i-1} + (2 + n^{-2})v_i - v_{i+1}\right) = 0,$$

which can be rewritten as

$$-d_{i-1} + (2 + n^{-2})d_i - d_{i+1} = 0, \qquad i = 2, \ldots, n-2.$$

From the first equation, we obtain

$$(1 + n^{-2})v_1 - v_2 = 0 \quad \Longrightarrow \quad (2 + n^{-2})d_1 - d_2 = 0.$$

From the last equation, we obtain

$$-v_{n-1} + (1 + n^{-2})v_n = 1 \quad \Longrightarrow \quad -d_{n-2} + (2 + n^{-2})d_{n-1} = 1.$$

Thus, the vector $\mathbf{d} = (d_1, \ldots, d_{n-1})^\top$ satisfies a tri-diagonal linear system

$$B'\mathbf{d} = \mathbf{e}_{n-1},$$

where $B' \in \mathbb{R}^{(n-1) \times (n-1)}$ is the symmetric tri-diagonal matrix

$$B' = \begin{bmatrix} 2 + n^{-2} & -1 & & \\ -1 & 2 + n^{-2} & -1 & \\ & \ddots & \ddots & \ddots \\ & & -1 & 2 + n^{-2} \end{bmatrix}.$$

The matrix $B'$ is strictly diagonally dominant with positive diagonal entries and nonpositive off-diagonal entries, hence an irreducible $M$-matrix (Berman & Plemmons, 1994). It is therefore positive definite and its inverse is entrywise nonnegative:

$$(B')^{-1} \geq 0 \quad \text{(entrywise).}$$

Since $\mathbf{d} = (B')^{-1}\mathbf{e}_{n-1}$, we obtain

$$d_i \geq 0, \qquad i = 1, \ldots, n-1.$$

Equivalently,

$$v_{i+1} - v_i = d_i \geq 0 \quad \Longrightarrow \quad v_1 \leq v_2 \leq \cdots \leq v_n.$$

The inequalities $S_{1,n} \leq S_{i,n} \leq S_{n,n}$ follow immediately from this monotonicity. $\qquad \square$

### B.2. Proof of Lemma 6

Note that the minimizers $(\mathbf{y}^{(i)})^*, i = 0, ..., T$ take the forms of

$$(\mathbf{y}^{(i)})^* = \underbrace{\frac{4n^2 + 1}{n^2} \left( \frac{1}{n^2} I_n + A \right)^{-1}}_{M} \mathbf{b}_x^{(i)}.$$

Combining Lemma 2 and Lemma 10, we have for all $i = 1, ..., n$,

$$4cn \leq M_{i,n} \leq 5Cn,$$

where $c = 1 - \frac{\pi^2}{12}$ and $C = 1 + \frac{\pi^2}{12}$. Thus, we have

$$\|(\mathbf{y}^{(i)})^*\|_\infty \leq 5Cn|x_i| \leq 5Cr_x\lambda < 10r_x\lambda < r_y\lambda,$$

which finishes the proof.

### B.3. Proof of Lemma 7

The proof of (a) is identical to the deterministic case. The proof of (b) follows the same reasoning as in Lemma 4. To establish (c), recall that

$$H_{sc}(\mathbf{x}) = \sum_{i=1}^{T} \frac{\lambda^2 L_f}{L} \left[ \Psi\left( -\frac{C_l}{\lambda} M_{n,n} x_{i-1} \right) \Phi\left( -\frac{C_r}{\lambda} M_{1,n} x_i \right) - \Psi\left( \frac{C_l}{\lambda} M_{n,n} x_{i-1} \right) \Phi\left( \frac{C_r}{\lambda} M_{1,n} x_i \right) \right].$$

Then one can verify that $\nabla^2 H_{sc}(\mathbf{x})$ is a tri-diagonal matrix whose entries are all of order $\mathcal{O}\left( \frac{L_f n^2}{L} \right)$. Consequently, $\left\| \nabla^2 H_{sc}(\mathbf{x}) \right\|_2 = \mathcal{O}\left( \frac{L_f n^2}{L} \right)$. To prove $(d)$, note that each coordinate of $\nabla_{\tilde{\mathbf{y}}} f_{sc}(\mathbf{x}; \tilde{\mathbf{y}})$ takes an order of $\mathcal{O}(\frac{\lambda L_f}{L})$, and hence $\|\nabla_{\tilde{\mathbf{y}}} f_{sc}(\mathbf{x}; \tilde{\mathbf{y}})\|_\infty = \mathcal{O}(\frac{\lambda L_f}{L})$.

For $\nabla_{\tilde{\mathbf{y}}} g_{sc}(\mathbf{x}; \tilde{\mathbf{y}})$, note that

$$\left\| \frac{\partial g_{sc}(\mathbf{x}; \tilde{\mathbf{y}})}{\partial \mathbf{y}^{(i)}} \right\|_\infty = \left\| \frac{L_g n^2}{4n^2 + 1} \left( \frac{1}{n^2} I_n + A \right) \mathbf{y}^{(i)} - L_g \mathbf{b}_x^{(i)} \right\|_\infty$$

$$\leq \frac{L_g}{4} \left\| \frac{1}{n^2} I_n + A \right\|_\infty \left\| \mathbf{y}^{(i)} \right\|_\infty + L_g |x_i|$$

$$\leq \frac{L_g}{4} \left( \frac{1}{n^2} + 4 \right) r_y \lambda + \frac{L_g r_x \lambda}{n}$$

$$\leq \frac{5}{4} L_g r_y \lambda + \frac{1}{10} L_g r_y \lambda \leq 2 L_g r_y \lambda,$$

which holds for all $i = 0, ..., T$. This implies that $\|\nabla_{\tilde{\mathbf{y}}} g_{sc}(\mathbf{x}; \tilde{\mathbf{y}})\|_\infty \leq 2L_g r_y \lambda$.

For $\|\nabla_{\mathbf{x}} g_{sc}(\mathbf{x}; \tilde{\mathbf{y}})\|_\infty$, note that

$$\left| \frac{\partial g_{sc}(\mathbf{x}; \tilde{\mathbf{y}})}{\partial x_i} \right| = L_g |y_n^{(i)}| \leq L_g r_y \lambda,$$

which yields that $\|\nabla_{\mathbf{x}} g_{sc}(\mathbf{x}; \tilde{\mathbf{y}})\|_\infty \leq L_g r_y \lambda$. Then, the proof is complete.

### B.4. Proof of Lemma 8

Note that $x_0 = \frac{\lambda}{C_l M_{n,n}} = \frac{\lambda}{\tilde{C}n}$. Since $|x_0| \geq \frac{\lambda}{\tilde{C}n}$ and $|x_i| < \frac{\lambda}{\tilde{C}n}$, we can find some $0 < j \leq i$ such that $|x_{j-1}| \geq \frac{\lambda}{\tilde{C}n}$ and $|x_j| < \frac{\lambda}{\tilde{C}n}$. Thus, look at

$$\frac{\partial H_{sc}(\mathbf{x})}{\partial x_j} = -\frac{\lambda L_f \tilde{C}n}{L} \left[ \Psi\left( -\frac{\tilde{C}n}{\lambda} x_{j-1} \right) \Phi'\left( -\frac{\tilde{C}n}{\lambda} x_j \right) + \Psi\left( \frac{\tilde{C}n}{\lambda} x_{j-1} \right) \Phi'\left( \frac{\tilde{C}n}{\lambda} x_j \right) \right]$$

$$- \frac{\lambda L_f \widetilde{C} n}{L} \left[ \Psi'\left(- \frac{\widetilde{C} n}{\lambda} x_j\right) \Phi\left(- \frac{\widetilde{C} n}{\lambda} x_{j+1}\right) + \Psi'\left(\frac{\widetilde{C} n}{\lambda} x_j\right) \Phi\left(\frac{\widetilde{C} n}{\lambda} x_{j+1}\right) \right]. \tag{15}$$

Then, if $\left|x_j - (1/L_h) \frac{\partial H_{sc}(\mathbf{x})}{\partial x_j}\right| \leq r_x \lambda / n$, then we have

$$L_h \|\mathcal{P}_{\mathcal{X}}[\mathbf{x} - (1/L_h) \nabla H_{sc}(\mathbf{x})] - \mathbf{x}\|_2 \geq \left|\frac{\partial H_{sc}(\mathbf{x})}{\partial x_j}\right| \geq \frac{\lambda L_f \widetilde{C} n}{L}.$$

Otherwise, i.e., $\left|x_j - (1/L_h) \frac{\partial H_{sc}(\mathbf{x})}{\partial x_j}\right| > r_x \lambda / n$, we have

$$\begin{aligned}
L_h \|\mathcal{P}_{\mathcal{X}}[\mathbf{x} - (1/L_h) \nabla H_{sc}(\mathbf{x})] - \mathbf{x}\|_2| &\geq L_h \left|\mathcal{P}_{\mathcal{C}^1_{r_x \lambda / n}} \left[x_j - (1/L_h) \frac{\partial H_{sc}(\mathbf{x})}{\partial x_j}\right] - x_j\right| \\
&\geq L_h \left(\frac{r_x \lambda}{n} - |x_j|\right) \geq L_h \left(\frac{r_x \lambda}{n} - \frac{\lambda}{\widetilde{C} n}\right) \\
&\overset{(i)}{\geq} \frac{c_0 n^2 L_f}{L} \left(r_x - \frac{1}{\widetilde{C}}\right) \frac{\lambda}{n} = \frac{c_0 L_f}{L} \left(r_x - \frac{1}{\widetilde{C}}\right) n\lambda,
\end{aligned}$$

where $(i)$ follows from $(c)$ of Lemma 7. Combining the above two cases completes the proof.

