# OpenReview forum: "Lower Complexity Bounds for Nonconvex-Strongly-Convex Bilevel Optimization with First-Order Oracles"
_ICML.cc/2026/Conference — ICML 2026 regular_

### Official Review · Reviewer_Nmi2 · 2026-03-12

**Soundness:** 3
**Presentation:** 3
**Significance:** 3
**Originality:** 3
**Overall Recommendation:** 5
**Confidence:** 5

**Summary:**

This paper considers nonconvex strongly convex bilevel optimization under deterministic and stochastic settings. The authors study the lower bound of the problem when one only has access to first-order oracles. In particular, they develop some novel hard instances on which the lower bounds of the bilevel algorithms are achieved in both deterministic and stochastic settings.

**Compliance With Llm Reviewing Policy:**

Affirmed.

**Key Questions For Authors:**

1. Although the lower bound study in theoretical computer science is mostly theory-driven. I am wondering if the authors can provide some empirical results on which the lower bounds are obtained? Even some simulation results would suffice and make the paper even stronger.

2. When we have access to second-order oracles like Jacobian-vector products, are there any existing/known results for the lower bounds, under similar smoothness assumptions? One paper I notice is [1], which seems to provide some analysis under certain smoothness and continuity assumptions.

3. Based on the lower bound analysis, could the authors provide some insights on the bilevel optimization algorithms design?


[1] Optimal Algorithms for Stochastic Bilevel Optimization under Relaxed Smoothness Conditions

**Limitations:**

This paper provides some solid analysis for understanding the lower bounds of bilevel optimization algorithms with first-order oracles. Some techniques may already exist in literature -- for example, zero-chain property. Also the problem setting requires strong convexity in the lower level problem, and it may require future efforts from the community to further analyze the general cases of bilevel optimization problems.

**Strengths And Weaknesses:**

Strengths:

1. Soundness. This paper is technically sound. The authors provide detailed analysis to characterize the lower bounds of bilevel optimization problems. Following the zero-chain property, they develop the proof outline as well as the hard instances to demonstrate the lower bounds of the problems.

2. Presentation. This paper is well written. The authors give a proof outline for the main theorems to help better understand the proof techniques used.

3. Significance and originality. Although the proof techniques exist in literature, applying it in bilevel problems is non-trivial.

Weakness:

1. There seems no numerical results to showcase the hardness of the problems.

---

> ### Author Rebuttal · Authors · 2026-03-27
>
> We thank the reviewer for the time and valuable comments!
>
> *Q. Although the lower bound study in theoretical computer science is mostly theory-driven. I am wondering if the authors can provide some empirical results on which the lower bounds are obtained? Even some simulation results would suffice and make the paper even stronger.*
>
> Good point! To further illustrate the behavior of our constructed worst-case instance, we have provided a numerical simulation (we cannot provide the figure here but will add it to the final revision).
> The results exhibit a clear two-phase behavior. In the early stage, the gradient norm is large but concentrated on a few coordinates, resulting in limited overall progress. As optimization proceeds, updates propagate sequentially across coordinates, reflecting the zero-chain structure of the construction and leading to delayed activation of later blocks. In the final stage, the iterates gradually stabilize, while the gradient norm decreases slowly, indicating diminishing global progress. Overall, these observations are consistent with the intuition behind our lower bound, where gradient information propagates inefficiently across blocks and thus slows convergence.
>
> *Q. When we have access to second-order oracles like Jacobian-vector products, are there any existing/known results for the lower bounds, under similar smoothness assumptions? One paper I notice is [1], which seems to provide some analysis under certain smoothness and continuity assumptions.*
>
> A. Great question! The work by Ji & Liang (Lower Bounds and Accelerated Algorithms for Bilevel Optimization, JMLR 2023) establishes lower bounds for second-order oracles, but focuses on the convex–strongly convex setting. Thanks for pointing out reference [1]. It is also a relevant work, though it considers a different (and more relaxed) smoothness condition than ours and does not provide a lower bound. We will cite this work. Deriving lower bounds under such relaxed smoothness conditions is an interesting direction for future study.
>
> *Q. Based on the lower bound analysis, could the authors provide some insights on the bilevel optimization algorithms design?*
>
> A. Our lower bound highlights several key insights for bilevel algorithm design. First, the construction shows that the main difficulty arises from the sequential dependency across blocks, where gradient information must propagate through the lower-level solution before affecting later coordinates. This implies that first-order methods inherently suffer from delayed progress, and accelerating inner optimization alone cannot remove this bottleneck.
>
> Second, the requirement that the upper-level gradient remains uniformly bounded independent of $\epsilon$ and $\mu$ indicates that the hardness is not due to ill-conditioning, but rather due to the structural coupling between levels. As a result, algorithms that rely on large or unstable hypergradients are unlikely to achieve optimal performance.
>
> Third, using second-order oracles could be a possible way to improve the upper and lower bounds. We want to leave this for future work.

---

> > ### Author Rebuttal · Reviewer_Nmi2 · 2026-04-04
> >
> > my concerns are addressed

---

### Official Review · Reviewer_5puq · 2026-03-12

**Soundness:** 3
**Presentation:** 3
**Significance:** 3
**Originality:** 2
**Overall Recommendation:** 4
**Confidence:** 3

**Summary:**

The paper studies oracle complexity lower bounds for smooth nonconvex-strongly-convex bilevel optimization with standard first-order oracles. Its main contribution is a hard-instance construction yielding deterministic and stochastic lower bounds that show a stronger dependence on the lower-level condition number $\kappa$ than in both smooth nonconvex single-level optimization and smooth nonconvex-strongly-concave min-max optimization. Concretely, the paper proves lower bounds of order $\Omega(\kappa^{3/2}\epsilon^{-2})$ in the deterministic setting and $\Omega(\kappa^{5/2}\epsilon^{-4})$ in the stochastic setting for zero-respecting first-order methods. Overall, the paper shows that nonconvex-strongly-convex bilevel optimization admits stronger lower bounds than these related settings under standard first-order oracle access.

**Compliance With Llm Reviewing Policy:**

Affirmed.

**Final Justification:**

My final recommendation remains **Weak Accept**. I view the paper as a meaningful theoretical contribution: it establishes stronger lower bounds, in their $\kappa$-dependence, for smooth nonconvex-strongly-convex bilevel optimization than those known for related smooth nonconvex and nonconvex-strongly-concave min-max settings. My main reservation remains the level of technical originality, since the proof relies substantially on adapting and combining known lower-bound ingredients. The rebuttal addressed my questions satisfactorily and reinforced, rather than changed, my overall assessment. In particular, it clarified the authors’ view of the technical novelty and their comparison with the concurrent work. Overall, I continue to think the paper clears the acceptance bar.

**Key Questions For Authors:**

1. Can the authors clarify more explicitly what they view as the main technical novelty relative to prior zero-chain based lower bounds and Carmon-style hard-instance constructions?

2. The comparison with the concurrent work of Chen & Zhang could be sharpened under matched assumptions and oracle models. In particular, the suggested choice $L_g=\mu$ is somewhat confusing, since it implies $\kappa=1$. A more explicit comparison under the same setting would help clarify the distinction between the two results.

3. How strongly does the proof strategy rely on lower-level strong convexity, and might some parts extend to the nonconvex-convex bilevel setting?

**Limitations:**

yes

**Strengths And Weaknesses:**

The main strength of the paper is that it addresses an important open theoretical question and derives lower bounds in the smooth nonconvex-strongly-convex bilevel setting that are strictly stronger, in their $\kappa$-dependence, than the corresponding lower bounds for smooth nonconvex optimization and smooth nonconvex-strongly-concave min-max optimization, in both the deterministic and stochastic settings. This is a meaningful contribution to the complexity theory of bilevel optimization. The paper is also generally clear and well organized.

The main weakness is in originality/technical novelty relative to prior lower-bound machinery. The proofs rely substantially on existing ingredients, especially the zero-chain argument and previously developed hard-instance building blocks, so the technical novelty comes primarily from their adaptation and combination in the bilevel setting rather than from a fundamentally new lower-bound technique. While this does not diminish the value of the result, the technical novelty lies mainly in the adaptation and combination of known ingredients.

---

> ### Author Rebuttal · Authors · 2026-03-27
>
> Thanks so much for the time and valuable comments!
>
> *Q. Can the authors clarify more explicitly what they view as the main technical novelty relative to prior zero-chain based lower bounds and Carmon-style hard-instance constructions?*
>
> A: Thank you for the question. Our construction indeed builds on the zero-chain framework and Carmon-style hard instances. The main technical novelty lies in how we adapt and extend these tools to the bilevel setting.
> Specifically, our key contribution is the design of the hard instances in Eqs. (5) and (10). This construction is nontrivial: we partition the variable $y$ into multiple blocks and carefully couple the end of each block with the beginning of the next to induce nonconvex interactions in the upper-level function (see Fig. 1). At the same time, these blocks are designed to preserve the zero-chain structure required for the strongly convex lower-level problem. To the best of our knowledge, this lower-level construction has not appeared in prior work. Moreover, another challenge in developing lower bounds for bilevel optimization is to ensure that the upper-level gradient remains uniformly bounded by a constant independent of $\epsilon$ and $\mu$. To the best of our knowledge, our construction is the first to achieve this while still yielding a nontrivial lower bound.
> While prior lower bounds for convex and nonconvex optimization rely heavily on zero-chain and related techniques, adapting these tools to obtain tight and meaningful lower bounds in new settings (such as bilevel optimization) is far from straightforward and requires careful, problem-specific design.
>
> *Q. The comparison with the concurrent work of Chen & Zhang could be sharpened under matched assumptions and oracle models. The suggested choice $L_g=\mu$ is somewhat confusing, since it implies $\kappa=1$. A more explicit comparison under the same setting would help clarify the distinction between the two results.*
>
> A. In Chen & Zhang, their construction of the lower-level instance (see eq. (14) therein) is $g(x,z)=\mu||z||^2 – L<x,z>$. Thus, this explicitly means that $\kappa=1$ in their case. Thus, we cannot adapt their bound to our setting, where we use general $L$ and any $\kappa>=1$. This is why we adapt our bound to their setting with $\kappa=1$, deriving a stronger bound than theirs in their setting.
>
> *Q. How strongly does the proof strategy rely on lower-level strong convexity, and might some parts extend to the nonconvex-convex bilevel setting?*
>
> A. For the convex case, some revisions are needed, such as adjusting the Laplacian matrix in eq. (3) and removing the identity matrix $I_n$ in eq. (5). This work (Ji&Liang, Lower Bounds and Accelerated Algorithms for Bilevel Optimization, JMLR 2023) can be useful here. The zero-chain properties should still apply. Overall, this extension seems quite feasible.

---

> > ### Author Rebuttal · Reviewer_5puq · 2026-04-02
> >
> > Thank you for the response. I maintain my positive evaluation of the paper.

---

### Official Review · Reviewer_VB7U · 2026-03-15

**Soundness:** 4
**Presentation:** 3
**Significance:** 4
**Originality:** 4
**Overall Recommendation:** 5
**Confidence:** 4

**Summary:**

This paper proves information-theoretic complexity lower bound for the smooth nonconvex-strongly-convex setup in bilevel optimization. Constructed lower bound instance requires at least $\Omega(\kappa^{3/2} \epsilon^{-2}$ and $\Omega(\kappa^{5/2} \epsilon^{-4})$ oracle calls for deterministic and stochastic case respectively, for any zero-respecting bilevel algorithms. This result further highlights the hardness of bilevel optimization compared to that of single-level and min-max optimization problems, as bilevel optimization subsumes the other two setups.

**Compliance With Llm Reviewing Policy:**

Affirmed.

**Final Justification:**

For this paper, my final recommendation is 'Accept'. I believe this paper clearly brings value to the optimization theory via complexity lower bound of bilevel optimization and authors were able to addressed the raised concerns.

**Key Questions For Authors:**

1. How does this result compare to the complexity lower bounds results with different choices of stochastic oracles or finite-sum setup?

**Limitations:**

Yes.

**Strengths And Weaknesses:**

This paper resolves one of the fundamental challenges in theoretical analysis of bilevel optimization with rigorous and fine-grained analysis. Meanwhile, it clearly demonstrates the intuition used in deriving the lower bound instances and lighten the connections to the existing literature. Not only ending up with proofs, this paper further guides the reader with potential application of the methodology and related open problems, adequately positioning this paper in the literature.

These are minor comments:
1. In definition 4, first-order oracle $O_g$ should be of the form $(g(x;y), \dots)$, not $(f(x;y), \dots)$.

---

> ### Author Rebuttal · Authors · 2026-03-27
>
> We thank the reviewer for the time and valuable comments!
>
> *Q: How does this result compare to the complexity lower bounds results with different choices of stochastic oracles or finite-sum setup?*
>
> A. Great question! We consider unbiased stochastic oracles with bounded variance $\sigma^2$. If we further assume mean-square smoothness of the stochastic gradients (i.e., the gradient estimators are Lipschitz in expectation), this enables sharper convergence characterizations, but typically requires variance-reduction techniques to achieve improved upper bounds.
>
> In the finite-sum setting, the noise is no longer captured by a uniform variance $\sigma^2$; instead, it depends on the heterogeneity among the $n$ component functions, often reflected through the variance of individual gradients around the full gradient.
>
> *Q. Typo on first-order oracle $O_g$:*
>
> A. Thanks so much! Will revise it.

---

> > ### Author Rebuttal · Reviewer_VB7U · 2026-04-04
> >
> > Thank you for the response, it fully resolved my concern. I am maintaining positive opinion the paper.

---

### Decision · Program_Chairs · 2026-04-30

**Decision:**

Accept (regular)

**Comment:**

All reviewers find that the theoretical results obtained in the manuscript are sufficiently novel and address an important question on the complexity of finding stationary points of nonconvex-strongly convex bilevel optimization problems. Please address the reviewers' comments when preparing the final version.